# Exploring Unconfirmed Transactions for Effective Bitcoin Address Clustering

## ABSTRACT

The advancement of clustering heuristics has demonstrated that the addresses of Bitcoin, which are protected by their anonymous mechanisms, can be de-anonymized. While the state-of-the-art (SOTA) clustering heuristics focus on confirmed transactions stored in the blockchain, they ignore unconfirmed transactions in the mempool. These unconfirmed transactions contain information about transactions before being stored in the blockchain, covering additional address associations that can improve Bitcoin address clustering.

In this paper, we bridge the gap by combining confirmed and unconfirmed transactions for effective Bitcoin address clustering. First, we introduce a reliable data collection framework to collect both confirmed and unconfirmed Bitcoin transactions. Second, we propose two novel clustering heuristics that exploit specific behavior patterns in unconfirmed transactions and uncover additional address associations. Finally, we construct a labeled dataset and experimentally show that the effectiveness of our proposed clustering heuristics, improving recall by at least three times compared to the SOTA clustering heuristics. Our findings demonstrate the value of unconfirmed transactions for Bitcoin address clustering and further reveal the challenges of achieving anonymity in cryptocurrencies. To the best of our knowledge, our study is the first to explore unconfirmed transactions for Bitcoin address clustering.

## CCS CONCEPTS

• **Security and privacy** → *Pseudonymity, anonymity and untraceability*.

## KEYWORDS

Bitcoin, address clustering, unconfirmed transactions, confirmed transactions, de-anonymization

## 1 INTRODUCTION

Introduced in 2008, Bitcoin [30] provides a pseudo-anonymous payment system that tries to decouple a user's real identity from his or her Bitcoin addresses. Bitcoin utilizes the blockchain as the distributed ledger. In Bitcoin, when a transaction is initiated, it is stored in the temporary storage of *unconfirmed* transactions, commonly referred to as the mempool. An *unconfirmed* transaction becomes a *confirmed* transaction only when stored in a confirmed block of the blockchain. Notably, a new confirmed transaction is considered irreversible when the blockchain receives six confirmed blocks after the transaction [4]. Furthermore, a portion of unconfirmed transactions may become *failed* transactions with no chance of being confirmed, due to either being replaced or paying insufficient fees.

In Bitcoin transactions, users employ a set of addresses to hide their real identities. That is, they can generate an unlimited number of new addresses for various transactions without revealing their real identities. The pseudo-anonymous ecosystem of Bitcoin has attracted an increasing number of users, including criminals who leverage Bitcoin to obfuscate their real identities during the transfer of illicit funds. To deal with such criminal activities in Bitcoin, a large number of studies focus on Bitcoin de-anonymization. Central to this domain is the notion of Bitcoin address clustering [50], which aims to identify multiple addresses controlled by the same entity, thus de-anonymizing these addresses. At present, the clustering heuristic stands as the predominant method for Bitcoin address clustering, which achieves address clustering by analyzing behavior patterns in confirmed transactions [17, 19, 28, 37, 41, 43]. For instance, transactions with multiple inputs usually arise when a user lacks an Unspent Transaction Output (UTXO) that has sufficient bitcoins to cover the payment. One straightforward idea, known as the *co-spend* heuristic [17, 28, 41], considers that all input addresses in a Bitcoin transaction belong to the same entity. In practice, clustering heuristics find extensive application in various domains, including case investigations [5, 9, 32] and the tracking of illicit funds [13, 24, 43], particularly within firms specializing in blockchain data analytics, such as Chainalysis [3].

However, the state-of-the-art (SOTA) clustering heuristics focus only on confirmed transactions but ignore unconfirmed transactions, leading to numerous undiscovered address associations. On the one hand, a portion of unconfirmed transactions will inevitably turn into failed transactions. Thus, the blockchain no longer contains information about these transactions. Consequently, focusing only on confirmed transactions results in the omission of potentially valuable address associations hidden in these failed transactions. On the other hand, unconfirmed transactions can provide several important insights into the state of transactions before being stored in the blockchain. For example, to incentivize miners to store a user's unconfirmed transaction in the blockchain more quickly, the user may initiate a new transaction with a high fee that spends the UTXO(s) of the unconfirmed transaction. This behavior forms a dependency chain in the mempool that contains address associations. However, the dependencies among transactions are not stored in the blockchain later. Thus, such insight cannot be captured by the analysis only based on confirmed transactions. Therefore, the comprehensive analysis of unconfirmed transactions can play an important role in Bitcoin address clustering.

In this paper, we present a practical approach for improving Bitcoin address clustering, leveraging both confirmed and unconfirmed transactions. First, we introduce a reliable data collection framework, including two sub-components: Confirmed Transaction Collector (CTC) and Unconfirmed Transaction Processor (UTP). Hereby, CTC, which utilizes a single node running a Bitcoin client *Bitcoin Core*[1], is responsible for copying confirmed transactions

*WWW, 2024, Singapore*
2023. ACM ISBN 978-x-xxxx-xxxx-x/YY/MM. . . $15.00
https://doi.org/10.1145/nnnnnnn.nnnnnnn

---

[1] https://bitcoin.org/en/releases/22.0/

in the blockchain. UTP is responsible for recording and processing all unconfirmed transactions in real time. It comprises five nodes, each of which runs a modified Bitcoin Core. Subsequently, we propose two novel clustering heuristics specifically designed for unconfirmed transactions, aiming to uncover additional address associations that are beyond the capabilities of the SOTA clustering heuristics. The principles of our proposed clustering heuristics are derived from the Replace-by-fee (RBF) proposed by Bitcoin Improvement Proposal (BIP)125 [11] and the unconfirmed transaction dependency chain mentioned in BIP141 [26]. Experimental results reveal the effectiveness of our approach in leveraging unconfirmed transactions to uncover address associations, significantly improving the clustering results of the SOTA clustering heuristics. To validate our approach, we construct a labeled dataset based on Bitcoin ordinal inscriptions [38] and demonstrate that our approach improves the recall with high precision by at least three times compared to the SOTA clustering heuristics. The increase in recall indicates that our approach uncovers additional address associations, thus reducing entities incorrectly clustered. Furthermore, we show that our approach reduces the number of entities in the clustering results of the SOTA clustering heuristics by at least 20.28%, which can reduce the error of addresses that should belong to the same entity, but being clustered into multiple entities. Finally, we find that unconfirmed transactions have a greater impact on the clustering results for future periods than those from past periods.

To the best of our knowledge, our study is the first to explore unconfirmed transactions to cluster addresses in Bitcoin. In summary, our main contributions in this paper are threefold:

- **Novel heuristics**: We propose two novel clustering heuristics to uncover additional address associations by analyzing the specific behavior patterns in unconfirmed transactions, in order to improve Bitcoin address clustering. Experimental results show that our proposed clustering heuristics can effectively utilize unconfirmed transactions to uncover address associations, significantly improving recall with high precision by at least three times.
- **Data collection**: We introduce a reliable data collection framework to record and process all unconfirmed transactions in Bitcoin in real time. We release a part of the dataset[2] as a benchmark for future studies.
- **Labeling method**: We present a method for constructing a labeled dataset based on Bitcoin ordinal inscriptions. This method addresses, to some extent, the critical issue in the field of Bitcoin address clustering, i.e., the lack of labeled datasets to validate clustering results. In this paper, we construct and release a dataset[3] encompassing 20 entities and 62,971 addresses.

## 2 BACKGROUND

### 2.1 Mempool in a Bitcoin Node

Bitcoin is established on a set of Bitcoin nodes, each of which stores a ledger of confirmed transactions. Bitcoin blockchain acts as the distributed ledger.

---

[2]See details at https://drive.google.com/drive/folders/1Vc5p9qro8zh6lV6lLqQMB4AtSvdhLjiT?usp=sharing

[3]See details at https://github.com/UnconfirmedTransactions/LabeledDataset

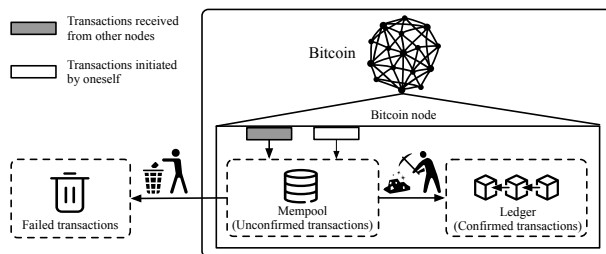

**Figure 1: Life-cycle of a Bitcoin transaction.**

As shown in Figure 1, before a transaction is stored in the blockchain, it (as an *unconfirmed* transaction) will be temporarily stored in the mempool of a node. The node will validate (or reject) the transaction that is from other nodes or initiated by itself based on established criteria. For instance, it validates the correctness of signatures. If the transaction meets the established criteria, the node will add the transaction to its mempool. Miners select transactions from their own mempools and then packaging these transactions to a block. They compete to append the block to the blockchain through proof-of-work. The winning miner propagates his or her block to other nodes, while other nodes append the block to their own ledgers. When the block is appended to the ledgers of all nodes, it indicates that the block is appended to the blockchain. Then, the transactions in the block become *confirmed* transactions. Note that a portion of unconfirmed transactions might never get confirmed due to, e.g., being replaced or paying insufficient fees. These transactions are considered as *failed* transactions.

Each unconfirmed transaction in the mempool has some additional fields, e.g., *replaceable*, *time*, *depends*, and *spentby*, which are not present in confirmed transactions. The latter three fields are exclusively available when a transaction is in Bitcoin mempool and disappear once the transaction is confirmed. (1) The field *replaceable* is a Boolean value, indicating whether this transaction can be replaced by another transaction. (2) The field *time* annotates the moment at which the transaction enters a particular node's mempool that may exhibit minor variances across different mempools. (3) The field *depends* of a transaction records unconfirmed transactions whose UTXO(s) is spent by this transaction. (4) The field *spentby* of a transaction records unconfirmed transactions spending outputs of this transaction. These fields contain rich information about a transaction before it is confirmed, which can be utilized in our study for Bitcoin address clustering.

### 2.2 Bitcoin Address Clustering

The SOTA clustering heuristics rely on behavior patterns in confirmed transactions to uncover address associations, which indicate whether these addresses are controlled by the same entity. For example, a common heuristic, known as the *co-spend* heuristic, considers that all inputs of a transaction are controlled by the same entity, because a valid transaction requires the signature of private keys corresponding to all inputs. In practice, however, a few studies [13, 17, 19] advocate the exclusion of Coinjoin transactions [27] prior to applying the *co-spend* heuristic. This is mainly because Coinjoin transactions employ a trustless method for combining

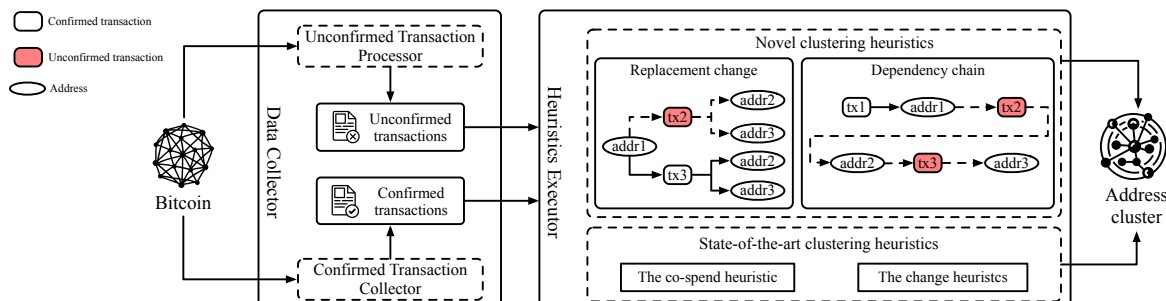

**Figure 2: Our approach for Bitcoin address clustering by combining unconfirmed transactions and confirmed transactions.**

multiple Bitcoin payments into a single transaction, thereby obfuscating the relationship between senders and recipients.

In addition, various heuristics, known as the *change* heuristics, have been introduced in previous studies [1, 6, 7, 19, 28]. In Bitcoin, an Unspent Transaction Output (UTXO) represents a certain amount of bitcoins. It is an indivisible unit and must be fully spent in a transaction. This results in the need for senders to use a change address to receive the remaining amount of bitcoins. Thus, the address of the sender (the input address) and the change address should be controlled by the same entity. For such heuristics, it is key to identify change addresses in transactions.

So far, the SOTA clustering heuristics focus only on behavior patterns in confirmed transactions that are already stored in the blockchain. However, they all ignore the additional information of a transaction before it is stored in the blockchain.

## 3 APPROACH

### 3.1 Overview

As shown in Figure 2, our approach consists of two components: Data Collector and Heuristics Executor. Data Collector contains two sub-components: Confirmed Transaction Collector (CTC) and Unconfirmed Transaction Processor (UTP). CTC copies confirmed transactions from the ledger of a node, while UTP collects and processes all unconfirmed transactions that appeared in mempools of deployed nodes. Data Collector subsequently transfers both confirmed and unconfirmed transactions to Heuristics Executor. Then, Heuristics Executor, consisting of our proposed clustering heuristics and the SOTA clustering heuristics, clusters Bitcoin addresses.

### 3.2 Data Collector

**CTC.** As shown in Figure 2, CTC with one node running a Bitcoin client, referred to as Bitcoin Core, copies confirmed transactions from the ledger of the node. In CTC, we optimize BlockSci [17], a widely used Bitcoin transaction parsing tool, as BlockSci-modified[4], which can parse Taproot addresses [46].

**UTP.** UTP consists of a set (five in this paper) of nodes, each of which runs a modified Bitcoin Core. It aims to collect as many unconfirmed transactions as possible in Bitcoin and continuously reconstruct the state of the Bitcoin mempool in each node. This

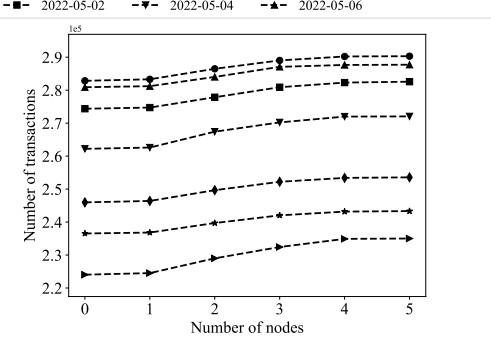

**Figure 3: The number of transactions collected vs. the number of nodes running a modified Bitcoin core.**

state hereby contains the details of each transaction in the mempool at a moment. Here, UTP tries to solve two key issues as follows:

(1) *How can UTP collect unconfirmed transactions in real time?* When using the Remote Procedure Call interface provided by Bitcoin Core to collect unconfirmed transactions, the first call has to obtain hashes of unconfirmed transactions in the mempool presently. Then, the second call retrieves detailed transactions based on these hashes. Due to the time gap between these two calls, part of the transactions may be removed from the mempool, resulting in missing these removed transactions. To collect unconfirmed transactions in real time, we modify Bitcoin Core. Our modified Bitcoin Core[5] can monitor the arrival of each transaction, then record transaction details from the moment it enters the mempool until it is confirmed or failed. The arrival of each unconfirmed transaction triggers UTP of a node for recording in real time to ensure that no unconfirmed transactions received by the node are missed.

(2) *How can UTP collect unconfirmed transactions in Bitcoin as many as possible?* Due to multiple factors such as the decentralized network of Bitcoin, network latency, and bandwidth limitations, unconfirmed transactions received by different nodes might vary. To achieve a comprehensive collection of unconfirmed transactions in Bitcoin, we perform experiments to evaluate the completeness of unconfirmed transactions collected by UTP. We increase the

---

[4]See details at https://github.com/UnconfirmedTransactions/BlockSci-modified

[5]See details at https://github.com/UnconfirmedTransactions/BitcoinCore-modified

**Table 1: Fields of an unconfirmed transaction.**

| Name | Content |
|---|---|
| txid | hash of transaction, not including witness data. |
| wtxid | hash of transaction, including witness data. |
| inputs | the inputs of transaction. |
| outputs | the outputs of transaction. |
| fee | transaction fee in BTC |
| vsize | virtual transaction size |
| weight | transaction weight |
| time* | local time when the transaction enters mempool |
| removetime* | local time when the transaction is removed |
| height* | block height when transaction enters mempool |
| descendantcount* | number of descendant transactions |
| descendantsize* | vsize of descendant transactions |
| descendantfees* | modified fees of descendant transactions |
| ancestorcount* | number of ancestor transactions |
| ancestorsize* | vsize of ancestor transactions |
| ancestorfees* | modified fees of ancestor transactions |
| depends* | unconfirmed transactions used as inputs |
| spentby* | unconfirmed transactions spending outputs |
| replaceable* | whether this transaction could be replaced |

$^1$ Fields with a star (*) are only present in unconfirmed transactions.

number of nodes and measure the number of deduplicated unconfirmed transactions collected per day from May 1, 2022 to May 7, 2022. Figure 3 shows that the number of deduplicated unconfirmed transactions rarely increases when the number of nodes reaches five. That is, UTP can collect approximately all unconfirmed transactions in Bitcoin when deploying five nodes. Therefore, we deploy five nodes, each of which runs a modified Bitcoin Core, to collect all unconfirmed transactions in Bitcoin. Table 1 shows the fields of an unconfirmed transaction in the mempool. In this paper, we focus on seven fields, i.e., *fee*, *vsize*, *time*, *removetime*, *depends*, *spentby*, and *replaceable*, which are relevant to subsequent analysis of behavior patterns in unconfirmed transactions. As shown in Table 1, there are additional fields that are not present in confirmed transactions, such as the field *ancestorcount*.

Finally, we build a mempool state database for the mempools of five nodes. In the database, we set *time* and *removetime* as indexes for each unconfirmed transaction. Given a specific time, the database is able to retrieve all unconfirmed transactions in each mempool at the moment. Note that a transaction output may be spent by multiple unconfirmed transactions. We make two adjustments to the original transaction structure. Specifically, the first field *output.is_spent*, a Boolean type, indicates whether the *output* has been spent by transactions. The second field *output.spent_tx* is a list containing hashes of transactions that spend this *output*.

Since failed transactions are removed from the mempool and no longer exist in Bitcoin, we can simply identify failed transactions by excluding confirmed transactions from unconfirmed transactions.

### 3.3 Novel Clustering Heuristics

We explore two mechanisms in unconfirmed transactions to design novel clustering heuristics.

**(1) Replace-by-fee (RBF)** [11]. It allows a sender to replace his or her unconfirmed transaction by initiating another transaction that pays a higher fee. Due to the limitation of fixed block size, miners give priority to transactions with a higher feerate (*fee*/*vsize*) to

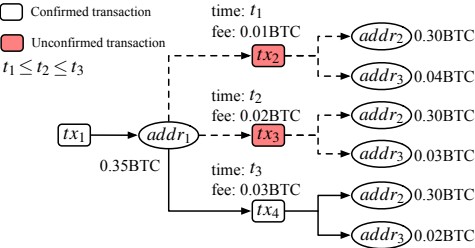

**Figure 4: Example of *replacement change* heuristic.**

maximize their profit. Note that a transaction can only be replaced when the field *replaceable* of the transaction is set to true.

**(2) Unconfirmed transaction dependency chain** [26]. The Bitcoin mempool is designed to accept unconfirmed transactions that spend UTXO(s) of other unconfirmed transactions. The user can initiate a new transaction to spend UTXO(s) of his or her unconfirmed transactions. As a result, it is common to form an unconfirmed transaction dependency chain in the mempool. In a dependency chain, each unconfirmed transaction spends UTXO(s) of the preceding unconfirmed transaction, which in turn spends UTXO(s) of another unconfirmed transaction, and so forth. Dependency chains typically form for two reasons. One is that users want miners to store their multiple transactions in the blockchain at one time, without waiting for a transaction to be confirmed before initiating a new one that spends UTXO(s) of the transaction. The other reason is related to a transaction pattern known as Child-Pays-for-Parent (CPFP). To incentivize miners to store the parent transaction of a user in the blockchain early, the user can initiate a child transaction that pays a high fee and spends UTXO(s) of the parent transaction.

For the first mechanism, we design a clustering heuristic, *replacement change* shown in Figure 2, to identify the change address.

**Replacement Change** heuristic. Let $T$ denote a set of transactions, defined as $T = \{tx_1, tx_2, \ldots, tx_n\}$ with $n \geq 2$. We identify the change address if (1) the field *replaceable* of each transaction in $T$ is True; (2) transactions in $T$ spend the same UTXO; (3) fees of transactions in $T$ increase as the field *time* increases; and (4) the address appears in the output of each transaction in $T$ and the amounts received by the address decrease as the field *time* increases.

This heuristic works for the following reason. In real-world trade of goods, the price of goods is typically negotiated between two parties and does not change arbitrarily. When the sender increases the fee, the amount paid to the recipient remains the same, while the amount received by the change address decreases. As shown in Figure 4, $tx_2$, $tx_3$, and $tx_4$ all attempt to spend the 0.35 BTC in $addr_1$. When the fee increases, the amount received by $addr_2$ remains the same, and the amount received by $addr_3$ decreases. Therefore, we can identify $addr_2$, which consistently receives the same amount, as the recipient address, and identify $addr_3$, which receives a gradually smaller amount, as the sender's change address.

For the second mechanism, we design another clustering heuristic, *dependency chain* shown in Figure 2, for the unconfirmed transaction dependency chain.

**Dependency Chain** heuristic. Let $T$ denote a sequence of unconfirmed transactions, defined as $T = \langle tx_1, tx_2, \ldots, tx_n \rangle$ with $n \geq 2$.

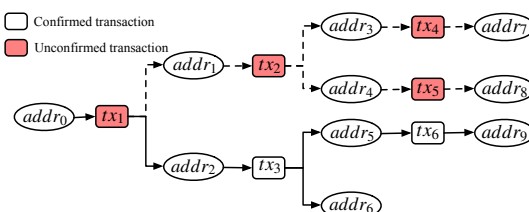

**Figure 5: Example of *dependency chain* heuristic.**

Each transaction $tx_k$ in the range $2 \leq k \leq n$ spends partial outputs of the previous transaction $tx_{k-1}$, denoted as $O_{k-1}$. Considering that $tx_1$ is the first transaction and spends outputs of confirmed transactions, we use $O_0$ to represent all inputs of $tx_1$. For every $k$ in the range $2 \leq k \leq n$, if (1) the hash of $tx_k$ is in the field *spentby* of $tx_{k-1}$; or (2) the hash of $tx_{k-1}$ is in the field *depends* of $tx_k$, all transactions in $T$ are initiated by one same entity, i.e., $\bigcup_{k=0}^{n} O_k$ are all controlled by the same entity.

This heuristic works for two reasons. First, in the design of popular Bitcoin wallets like Binance, Coinbase, Electrum, and BlueWallet, users can view and spend UTXO(s) of unconfirmed transactions initiated by themselves. To protect funds for users, these wallets do not allow users to view and spend UTXO(s) sent to them in the unconfirmed transactions initiated by others. Second, a Bitcoin transaction is considered irreversible after the blockchain receives six new blocks. Therefore, for commercial trades, users do not use unconfirmed transactions as an indicator of fund arrival and do not spend UTXO(s) of unconfirmed transactions. Instead, they have to wait for new blocks to ensure the security and irreversibility of the transactions. As a result, transactions between different entities do not form an unconfirmed transaction dependency chain.

As shown in Figure 5, we can identify two dependency chains: $\langle tx_1, tx_2, tx_4 \rangle$ and $\langle tx_1, tx_2, tx_5 \rangle$, while the three transactions $tx_1$, $tx_3$ and $tx_6$ and the two transactions $tx_1$ and $tx_3$ do not form a dependency chain. Applying this new heuristic, we can cluster $\{addr_0, addr_1, addr_3\}$ into the first entity and $\{addr_0, addr_1, addr_4\}$ into a second entity. These two entities can be further merged into a larger entity due to the common address $addr_0$ in both clusters.

## 4 EVALUATION

To evaluate the effectiveness of our approach, this section presents the experimental results to address three key issues as follows:

(1) **Clustering result validation.** Is the clustering result of our approach accurate, and does our approach possess the capability to uncover additional address associations? (Section 4.3)

(2) **Impact measurement.** How much impact does our approach have on the clustering results of the SOTA clustering heuristics? (Section 4.4)

(3) **Temporal analysis.** What pattern does the impact of our approach show across different periods? (Section 4.5)

### 4.1 Dataset

We collect a total of 116,514,258 unconfirmed transactions (from May 1, 2022 to May 31, 2023). Among these, 113,296,795 (97.24%) unconfirmed transactions become confirmed transactions, while the

rest of them become failed transactions. These confirmed transactions involve 179,352,220 Bitcoin addresses, while the failed transactions involve 12,366,745 Bitcoin addresses, 843,892 of which are not recorded in any confirmed transactions. As shown in Appendix A, we perform two case studies on these unconfirmed transactions. Both demonstrate the presence of behavior patterns in unconfirmed transactions, while not present in confirmed transactions.

### 4.2 Baseline

As shown in Figure 2, we employ six SOTA clustering heuristics as the baseline.

**Co-spend** (short for CS) considers all inputs of a transaction are controlled by the same entity if the transaction is not a Coinjoin transaction. We use the algorithm developed by Goldfeder *et al.* [7] to determine whether a transaction is a Coinjoin transaction.

**Androulaki *et al.*** (short for A) [1] identify the change address of a transaction sender if (1) the transaction must have exactly two outputs; and (2) the address is the only *fresh* address in the outputs, meaning that it has not been previously used in the blockchain.

**Meiklejohn *et al.*** (short for M) [28] identify the change address of a transaction sender if (1) the transaction is not a coinbase transaction; (2) the address is the only *fresh* address in the outputs; and (3) there is no address used as both an input and an output in this transaction.

**Goldfeder *et al.*** (short for G) [7] utilize the criteria established by Meiklejohn *et al.* [28], but they also add a further condition: (4) the transaction cannot be a Coinjoin transaction.

**Ermilov *et al.*** (short for E) [6] identify the change address of a transaction sender if (1) the number of inputs is not two; (2) the transaction has exactly two outputs; (3) there is no address used both as an input and an output in the transaction; (4) the address is the only *fresh* address in outputs; and (5) the amount received by the address is precise to a minimum of four decimal places.

**Kappos *et al.*** (short for K) [19] identify the change address of a transaction sender if (1) the transaction is a node in a *peel chain*; (2) the amount received by the address is spent and the spent transaction is also a node in the *peel chain*.

To describe experimental results clearly, we assign a name for each clustering result, composed of the heuristic and the state of transactions. The SC clustering result refers to the result of applying one of the SOTA clustering heuristics to confirmed transactions. The SF clustering result refers to the result of applying one of the SOTA clustering heuristics to failed transactions. The NU clustering result refers to the result of applying our proposed clustering heuristics to unconfirmed transactions. We denote the merging of clustering results with a plus sign. For instance, SC+SF represents a merge of SC and SF clustering results under the same heuristic. In cases where no clustering heuristic is applied, we refer to it as None, with each address being considered as an isolated entity.

### 4.3 Clustering Result Validation

To validate clustering results, we construct a labeled dataset and analyze the clustering results from multiple metrics.

**Labeling method.** Validating clustering results requires the availability of labeled datasets. However, there is no publicly available

labeled dataset since Bitcoin is pseudo-anonymous, and the transactions we analyze are relatively recent. Thus, we propose a method for constructing a labeled dataset to validate our clustering results.

The labeling method is based on Bitcoin ordinal inscriptions [44]. Bitcoin ordinal inscriptions are digital assets created by attaching information to an individual satoshi, the smallest denomination in Bitcoin, through the Ordinals protocol [38]. Two features are worth noting in this protocol. First, creating an individual ordinal inscription must follow a two-phase procedure: a commit transaction and a reveal transaction [38]. Thus, both commit and reveal transactions are initiated by the same entity. Second, an ordinal inscription collection consists of a set of individual ordinal inscriptions created by an artist or a group of artists. The parent-child inscription mechanism is utilized to create a collection, with child inscriptions being created exclusively by the owner of the parent inscription, resulting in all children being members of the same collection [38]. Thus, all ordinal inscriptions of a collection are created by the same entity. In summary, the input addresses of both commit and reveal transactions for each inscription in a collection are controlled by the same entity (see more details in Appendix B.1).

The specific process of the labeling method is given as follows. First, we gather a Bitcoin ordinal inscription collection, defined as $S = \{o_1, o_2, \ldots, o_n\}$ with $n \geq 2$. For each ordinal inscription $o_k$ in $S$, we identify its corresponding commit transaction $ctx_k$ and reveal transaction $rtx_k$. Next, we extract the input addresses of transaction $ctx_k$ and $rtx_k$ in the range $1 \leq k \leq n$, denoted as $I_k$. Finally, we consider $\bigcup_{k=1}^n I_k$ are controlled by the same entity.

For this paper, we gather 20 collections (entities) from the website [33] and label 62,971 addresses as the validation dataset. Specific details of the dataset are described in Appendix B.2.

**Validation metrics.** We measure clustering results from two aspects. First, we show the number of entities successfully identified ($N$). Second, we evaluate the quality of addresses in each identified entity through four metrics: Precision ($P$), Recall ($R$), Weighted Precision ($WP$), and Weighted Recall ($WR$). The first two metrics are commonly used in the study [2], while the last two metrics are introduced in the study [43]. The definitions of these four metrics are as follows, where $m$ denotes the total number of entities, i.e., 20, and $E_i$ denotes $i$th entity. Addresses of $E_i$ are clustered into $n$ clusters, with $c_{ij}$ representing the $j$th cluster of $E_i$. We denote the union of these clusters as $C_i$. We use the set $v_{ij}$ to denote the addresses of $Ei$ that are clustered into the cluster $c_{ij}$. We denote the union of $v_{ij}$ as $V_i$. $w_{ij}$ represents the proportion of the set $c_{ij}$ within entity $E_i$. The greater the number of addresses within a cluster, the more accurately it reflects the characteristics of the entity and, thus, the higher its significance in the clustering results.

$$P = \frac{\sum_{i=1}^m |V_i|}{\sum_{i=1}^m |C_i|}, \quad R = \frac{\sum_{i=1}^m |V_i|}{\sum_{i=1}^m |E_i|}$$

$$WP = \frac{1}{m} \sum_{i=1}^m \sum_{j=1}^n w_{ij} \frac{|v_{ij}|}{|C_i|}, \quad WR = \frac{1}{m} \sum_{i=1}^m \sum_{j=1}^n w_{ij} \frac{|v_{ij}|}{|E_i|}$$

$$\text{where } v_{ij} = E_i \cap c_{ij}, \ V_i = \bigcup_{j=1}^n v_{ij}, \ C_i = \bigcup_{j=1}^n c_{ij}, \ w_{ij} = \frac{|c_{ij}|}{|C_i|}.$$

**Validation results.** Table 2 demonstrates the effectiveness of our proposed clustering heuristics compared to the SOTA clustering heuristics. Our proposed clustering heuristics identify more entities

**Table 2: Comparison between our proposed clustering heuristics and SOTA clustering heuristics.**

| Heuristics | N | P(%) | R(%) | WP(%) | WR(%) |
|---|---|---|---|---|---|
| CS | 8 | 0.09 | **4.47** | 7.37 | 2.81 |
| CS+NU | 18 | 94.27 | 74.32 | 42.23 | 18.73 |
| CS+A | 10 | 28.49 | **3.93** | 14.11 | 2.85 |
| CS+A+NU | 16 | 94.49 | 59.58 | 35.53 | 18.24 |
| CS+M | 12 | 55.29 | **11.84** | 11.75 | 0.24 |
| CS+M+NU | 15 | 90.29 | 52.04 | 27.89 | 6.23 |
| CS+G | 12 | 55.29 | **11.84** | 11.75 | 0.24 |
| CS+G+NU | 15 | 90.29 | 52.04 | 27.89 | 6.23 |
| CS+E | 9 | 35.43 | **4.96** | 11.96 | 2.81 |
| CS+E+NU | 18 | 94.58 | 73.87 | 42.15 | 18.74 |
| CS+K | 12 | 75.21 | **12.46** | 12.52 | 8.32 |
| CS+K+NU | 16 | 90.43 | **77.53** | 27.72 | 16.28 |

while achieving a precision of over 90%. Notably, our proposed heuristic significantly improves recall. Even in the cases of CS+M and CS+G, where the improvement is the smallest, our proposed clustering heuristics still improve recall by three times. This indicates that our proposed clustering heuristics can uncover many additional address associations that are beyond the scope of the SOTA clustering heuristics. Both weighted precision and weighted recall exhibit a significant improvement, further showing our proposed clustering heuristics can uncover additional address associations and identify more addresses belonging to the same entity. Significantly, the results for both CS+M and CS+G are identical. This is because CS already achieves the exclusion of Coinjoin transactions, which is the sole distinction between CS+G and CS+M.

## 4.4 Impact Measurement

Building upon the demonstrated effectiveness of our proposed clustering heuristics in Section 4.3, we proceed with measuring their impact on the SC clustering results.

**Settings.** We apply the SOTA clustering heuristics to failed transactions and our proposed clustering heuristics to unconfirmed transactions to uncover additional address associations. When no clustering heuristic is applied, we consider each address in the confirmed transactions between May 1, 2022 and May 31, 2023 as an isolated entity, resulting in a total count of 179,352,220 isolated entities.

**Measurement metric.** We employ the reduced number of entities in the clustering results as the metric to measure the impact of our approach. Suppose the SOTA clustering heuristics cluster addresses of an entity into $n$ clusters, denoted as $\{C_1, C_2, \ldots, C_n\}$ with $n \geq 2$. Our proposed clustering heuristics produce an additional cluster, denoted as $C_{n+1}$. If $C_{n+1} \cap C_i \neq \emptyset$ for $i$ in the range $1 \leq i \leq n$, $C_{n+1}$ can merge these $n$ clusters, resulting in a reduced number of entities. It is worth noticing that $C_i$ may contain only one Bitcoin address for $i$ in the range $1 \leq i \leq n$. Therefore, the reduction in the number of entities reflects the ability of our approach to reduce the error of addresses that should belong to the same entity being clustered into multiple entities.

**Measurement results.** Figure 6(a) shows the impact of failed transactions on the SC clustering results. The SC+SF clustering result has a reduced number of entities compared to the SC clustering result across various SOTA clustering heuristics. Notably, the most

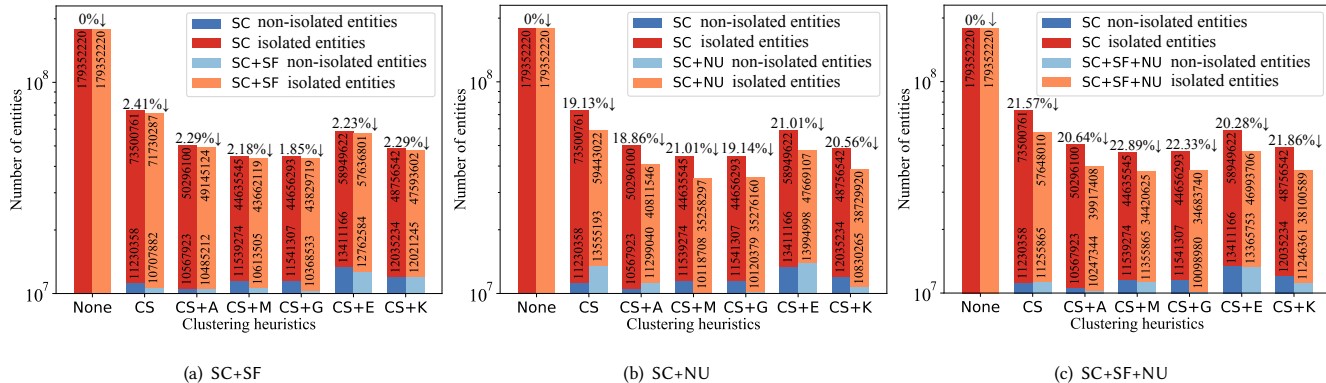

(a) SC+SF                                      (b) SC+NU                                      (c) SC+SF+NU

**Figure 6: Comparison of the number of entities in the** SC **clustering result and other three clustering results. While an isolated entity contains only one Bitcoin address, a non-isolated entity contains multiple Bitcoin addresses.**

significant reduction is observed in the result of the CS heuristic, with a reduction of 1,770,474 entities (2.41% of the total entities). This highlights that failed transactions contain additional address associations that can be uncovered by the SOTA clustering heuristics but are currently ignored.

Figure 6(b) shows the impact of our proposed clustering heuristics on the SC clustering results. The SC+NU clustering result has significantly fewer entities than the SC clustering result across various clustering heuristics. Notably, the CS+M clustering result is the most affected, with a reduction of 9,377,248 entities (21.01% of the total entities). This indicates that our proposed clustering heuristics reveal numerous address associations in unconfirmed transactions that are beyond the scope of the SOTA clustering heuristics.

Figure 6(c) shows the comprehensive impact of both failed transactions and our proposed clustering heuristics on the SC clustering results. The results indicate that a portion of the SF clustering results and NU clustering results exhibit no overlap, further reducing the number of entities. The CS+M clustering result is the most affected, with a reduction of 10,214,920 entities (22.89% of the total entities). Our approach utilizes failed transactions and our proposed clustering heuristics to uncover numerous additional address associations, significantly improving the SC clustering results.

## 4.5 Temporal Analysis

Considering that the state of a transaction changes over time, and UTP and CTC capture distinct state information about the same transaction at different moments, our approach has varying degrees of impact on the SC clustering results across different periods. In the following, we conduct a monthly temporal analysis of our approach's impact on the SC clustering results in different periods. Moreover, we also measure this impact by the reduced number of entities in the clustering results.

First, as shown in Figure 7, our approach has the most significant impact on the SC clustering results in the current month, and the impact on other months decreases month by month. Notably, we observe that the clustering results of our approach in the current month have a more significant impact on the SC clustering results in the subsequent month compared to the previous month. This

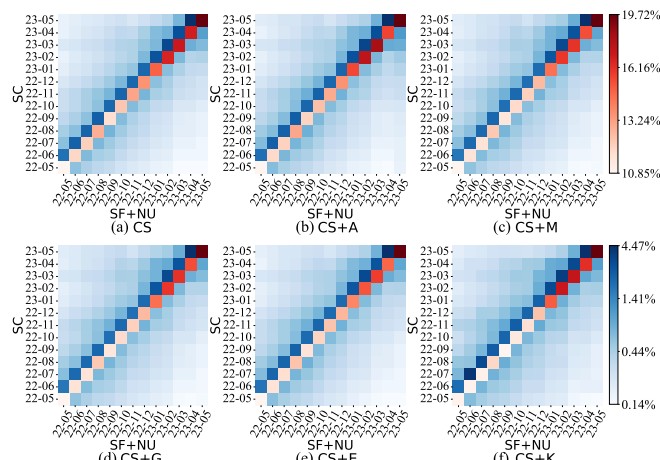

(a) CS                    (b) CS+A                    (c) CS+M

(d) CS+G                    (e) CS+E                    (f) CS+K

**Figure 7: Temporal impact of our approach on the SC clustering results.**

is primarily attributed to the fact that unconfirmed transactions collected by UTP in the current month may be confirmed in the subsequent month, thus leading to a more significant impact on the SC clustering results in the subsequent month.

Second, as shown in the diagonal, the impact of our approach exhibits a growing trend over time, indicating its enduring effect. Notably, the impact experiences a significant enhancement in January 2023. Our analysis attributes this phenomenon primarily to the emergence and widespread adoption of Bitcoin ordinal inscriptions in January 2023 [38]. When users create a collection of ordinal inscriptions, they often utilize unconfirmed transaction dependency chains to create numerous ordinal inscriptions at the same time.

## 5 DISCUSSION

**The impact of Coinjoin transactions on clustering heuristics.** Coinjoin transactions render the SOTA clustering heuristics ineffective. However, our proposed clustering heuristics utilize replacement transactions and unconfirmed transaction dependency

chains. To ensure timely transaction confirmation, users avoid employing Coinjoin transactions to facilitate operations in both cases. Furthermore, the *replacement transaction* heuristic identifies the change address definitely, unlike the SOTA clustering heuristics, which may produce false positives.

**False positive of our proposed clustering heuristics.** The *dependency chain* heuristic may produce false positives under the following specific situation. To accelerate the confirmation of a transaction, the sender transmits the transaction hash and the corresponding UTXO to the recipient before the transaction is confirmed. The recipient then utilizes this UTXO to initiate a new transaction, forming an unconfirmed transaction dependency chain. In this dependency chain, the Bitcoin addresses involved are not controlled by the same entity. However, this situation remains infrequent due to its potential association with unconfirmed transaction attacks, a subtype of double-spend attacks.

**Labeled dataset construction.** With the development of third-party platforms (no-code inscription tools) for Bitcoin ordinal inscriptions, users often use these platforms to create ordinal inscription collections for convenience. In this situation, a third-party platform creates multiple collections on behalf of users. Consequently, all input addresses for the ordinal inscription creation transactions in these collections are controlled by the third-party platform. While this situation introduces certain imperfections into the labeled dataset we construct, the address associations within each entity in our dataset remain accurate. Furthermore, a significant portion of ordinal inscription collections in our dataset are created in the early days when Bitcoin ordinal inscriptions are prevalent, predating the development of third-party platforms. Therefore, our dataset is minimally affected by this situation.

**User privacy leakage in unconfirmed transactions.** The experimental results in Section 4 reveal that unconfirmed transactions can significantly reduce the anonymity of Bitcoin, but it ultimately benefits Bitcoin users by motivating further research into privacy protocols for the mempool. Individuals who are worried about protecting their privacy may opt to cryptocurrencies that prioritize privacy, such as Zcash. However, prior studies demonstrate that even these cryptocurrencies do not guarantee complete anonymity [18, 22].

## 6 RELATED WORK

### 6.1 Clustering Bitcoin Addresses

Many studies attempt to achieve de-anonymization by proposing various clustering heuristics. The SOTA clustering heuristics can generally be categorized into two main groups: the co-spend heuristic and the change heuristic. The co-spend heuristic, observed in the white paper [30], is applied in many studies [1, 23, 28, 37, 41]. On the basis of the co-spend heuristic, Kalodner *et al.* [17] propose to reduce clustering interference caused by Coinjoin transactions. Then, Meiklejohn *et al.* [28] and Androulaki *et al.* [1] propose the change heuristic to determine which transaction output is the address to receive change. Goldfeder *et al.* [7] and Ermilov *et al.* [6] further refine this heuristic. The change heuristic has been used in multiple studies to track illicit fund flows [13, 14, 24, 34–36, 47]. Recently, Kappos *et al.* [19] consider the transaction pattern *peel chain* to identify the change address.

There are also studies on analyzing the effectiveness of various clustering heuristics [2, 8, 12, 25, 31, 50]. Cazabet *et al.* [2] highlight that only employing the co-spend heuristic has a relatively low recall but a high precision. Nick *et al.* [31] assess the accuracy of various clustering heuristics using a ground-truth dataset and find that, on average, over 69% of an entity's addresses could be successfully clustered using only the co-spend heuristic. Zheng *et al.* [50] demonstrate that the existing clustering heuristics do not guarantee the comprehensiveness, accuracy, and efficiency of the clustering results. Liu *et al.* [25] point out that all clustering heuristics rely on confirmed transactions stored in the blockchain. In this paper, we propose two clustering heuristics to uncover additional address associations in unconfirmed transactions.

### 6.2 Analyzing Bitcoin Mempool

Related studies focus on two issues: predicting the transaction confirmation time and analyzing unconfirmed transactions.

**Predicting the transaction confirmation time.** Many studies [10, 21, 29, 42, 48, 49] propose various methods to estimate the confirmation time. Gundlach *et al.* [10] predict the confirmation time of Bitcoin transactions by modeling the confirmation time as the time to ruin of a Cramer-Lundberg (CL) model. Ko *et al.* [21] and Zhang *et al.* [48] employ machine learning techniques to predict confirmation time of unconfirmed transactions, taking into account various factors such as the confirmation time of historical transactions, block states, and mempool states.

**Analyzing unconfirmed transactions.** Saad *et al.* [39, 40] investigate the impact of DDoS attacks on the mempool size and the fees paid by users. Meanwhile, Dae-Yong *et al.* [20] examine the variation of unconfirmed transactions in different mempools through the Jaccard similarity index. They find that unconfirmed transactions in mempools are significantly different when a new block is produced. Kallurkar *et al.* [16] focus on statistics of failed transactions and the primary reasons for the transaction failure. Furthermore, they point out that the area of failed transactions remains unexplored. To further explore the impact of the mempool on users, we focus on the user privacy disclosed by unconfirmed transactions (including failed transactions) in the mempool, and design clustering heuristics for unconfirmed transactions.

## 7 CONCLUSION AND FUTURE WORK

In this paper, we present a practical approach to cluster Bitcoin addresses by combining confirmed and unconfirmed transactions, significantly improving Bitcoin address clustering. The key idea is to explore specific behavior patterns in unconfirmed transactions and propose two novel clustering heuristics for unconfirmed transactions. Then, we construct a labeled dataset based on Bitcoin ordinal inscription to validate the clustering result, and measure the impact of our approach. Experimental results reveal that our proposed clustering heuristics can uncover additional address associations and reduce the error of addresses controlled by the same entity being clustered into multiple entities.

In future, we aim to extend our analysis to other cryptocurrencies based on the UTXO model, such as Litecoin and Dogecoin. We will also analyze the Ethereum mempool and use the mempool data to explore the traceability of funds under the account-balance model.

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

# APPENDIX

## A  CASE STUDIES

Although unconfirmed transactions or even failed transactions are not stored in the blockchain, they still reveal the motivations behind why users initiate these transactions. This information is highly valuable for analyzing transaction behaviors. In this section, we demonstrate the utility of failed transactions in analyzing transaction behaviors through two case studies.

### A.1  Binance Exchange Address Analysis

One of the main safeguards used by Bitcoin exchanges to prevent attacks is the use of cold and hot storage technology, with carefully designed risk control systems [23].

There are three main types of addresses controlled by Bitcoin exchanges: hot wallet addresses, cold wallet addresses, and user wallet addresses. The primary function of the hot wallet is to maintain a bitcoin pool for the user's withdrawal demand. The user wallet address is created by the Bitcoin exchange and the private key of the address is held by the Bitcoin exchange. Users can deposit bitcoins to Bitcoin exchange by the user wallet address.

Figure 8 shows a failed transaction related to the Binance exchange, a confirmed transaction to replace the failed transaction, and another two confirmed transactions related to Binance exchange. The address *bc1q...7s3h* is labeled with the Binance exchange by Blockchain.com [15]. When the transaction *66c1...9dc6* is confirmed, the transaction *dc43...ef51* is initiated to transfer bitcoins in the address *bc1q...wwvq* to Binance exchange address. Three minutes after the transaction *dc43...ef51* is initiated, the same user initiates the transaction *4161... bb25*. The second transaction spends the same UTXO as the first transaction but pays a much higher fee. As a result, miners choose the second transaction to pack into a block that is eventually confirmed. The transaction *dc43...ef51* turns out to be a failed transaction that is removed by all Bitcoin nodes and can never be confirmed again. Then, another transaction *6903...4583* is initiated with 100 inputs and 1 output, which transfers bitcoins in the address *19Fa...Hd5X* to the address *bc1q...7s3h*.

From the content of the transaction *dc43...ef51* and transaction *6903...4583*, the purpose of the address *bc1q...wwvq* is to transfer its bitcoins to the address *bc1q...7s3h*. A question worth analyzing is why the user would replace the transaction *dc43...ef51* with the second one *4161...bb25*.

To answer this question, we investigate 127 confirmed transactions involving the address *bc1q...wwvq*. All outgoing transactions of the address *bc1q...wwvq* consist of one input and one output. When the address *bc1q...wwvq* transfers bitcoins to other addresses, the recipient address is always the address *19Fa...Hd5X*. Then, the address *19Fa...Hd5X* transfer bitcoins to the address *bc1q...7s3h*. Both incoming and outgoing transactions of the address *bc1q...wwvq* occur in pairs, i.e., the address receives bitcoins and then transfers out all the bitcoins it receives. Therefore, we infer that the transaction *dc43...ef5* was a mistake by the Binance exchange. Address *bc1q...7s3h* is a hot wallet address of Binance exchange, and under the rules of Binance exchange, address *bc1q* cannot transfer bitcoins directly to address *bc1q...7s3h*. Therefore, when the transaction *dc43...ef51* was initiated, the operator or script discovered the misoperation and initiated another high-fee transaction *4161...bb25* to replace the first one.

As seen from the above discussion, the addresses controlled by the Binance exchange have distinct roles and perform specific responsibilities. The Binance exchange carefully plans the transaction relationships among addresses to prevent the casual transfer of bitcoins among them.

Through analysis of unconfirmed transactions, we can shed light on the Binance exchange's internal risk prevention and control mechanisms, and can assist regulators in verifying the cryptocurrency exchange's reported information and inadvertently disclosed transfer behavior.

### A.2  Potential Dust Attacks Against Whale Addresses

The dust attack is defined as malicious behavior targeting Bitcoin users and privacy by sending tiny amounts of bitcoins to victims' addresses [45]. The aim of the dust attacker is to reveal the user's identity by collecting information on where these tiny amounts are combined when the user initiates a new transaction through their cryptocurrency wallet software. The attackers track the transaction activity of these addresses in an attempt to link the dusted addresses and identify the person or company behind them [45].

Figure 9 shows a potential dust attack against whale addresses found in a failed transaction. Figure 9 contains two confirmed transactions and one failed transaction. In this scenario, the transaction *3180...a362* is first initiated with two inputs and two outputs. When this transaction is confirmed, the address *1KqX...JYEQ* transfers its received bitcoins to the address *1FU6...8hKf* through the transaction *4f6e...9ce3*. We cannot find anything unusual about this address *1KqX...JYEQ* from confirmed transactions.

However, before the transaction *4f6e...9ce3* is initiated, the transaction *f125...6b9a* is first initiated. These two transactions spend the same UTXO *1KqX...JYEQ*. Due to the much higher fee of the transaction *4f6e...9ce3*, miners choose the transaction *4f6e...9ce3* to package into a block that eventually is confirmed. Therefore, the transaction *f125...6b9a* turns out to be a failed transaction. However, the failed transaction *f125...6b9a* reflects the malicious behavior of the user that is not presented in confirmed transactions.

More specifically, the transaction *f125...6b9a* has one input and 9 outputs. It is noteworthy that all 8 outputs of this transaction have the same revenue, i.e. 0.00000666 BTC ($ 0.12). The remaining

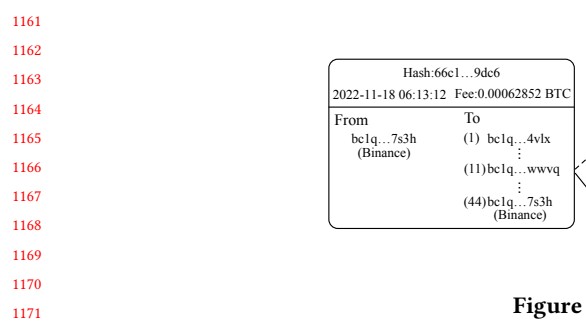

Figure 8: Binance exchange address analysis.

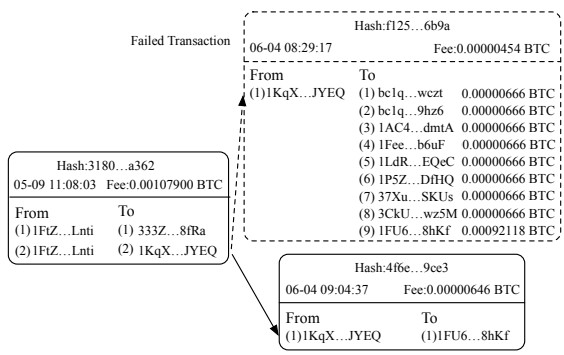

Figure 9: Potential dust attack against whale addresses.

output is the same as the sender and is the change address of the sender. The address *bc1q...wczt*, *bc1q...9hz6* and *1Fee...b6uF* are labeled with *FBI3*, *FBI 2 (Silk Road)* and *MtGox Hacker* respectively by Blockchain.com [15]. The stolen bitcoins of the Bitfinex Hack 2016 continue to converge to the address *bc1q...wczt* that currently has a balance of more than 94,643 BTC without any transfers out. Nearly 70,000 BTC confiscated by the U.S. government from the black market site *Silk Road* are transferred to the address *bc1q...9hz6* in early November, 2020. As of this writing, the bitcoins have not been moved or liquidated. Former Mt.Gox CEO Mark Karpeles confirmed that the bitcoins residing at the address *1Fee...b6uF* were stolen from the Mt.Gox exchange. The address *1P5Z...DfHQ* is controlled by the FTX exchange which declared bankruptcy on November 14, 2022. The other addresses except *1FU6...8hKf* also have a large balance when this failed transaction *f125...6b9a* is initiated. Currently, the balance of the address *3CkU...wz5M* is 0. But, the balance of the address *3CkU...wz5* is more than 50,620 BTC when the failed transaction *f125...6b9a* is initiated. Therefore, the address *1KqX...JYEQ* sends a very small amount of bitcoins to multiple whale addresses via the transaction *f125...6b9a*, with the purpose of a dust attack.

Focusing only on confirmed transactions in the blockchain fails to uncover hidden users' intentions. On the one hand, unconfirmed transactions can also be used to identify malicious behavior that cannot be identified by confirmed transactions alone. On the other hand, unconfirmed transactions allow for the identification of malicious behavior before it takes place.

Table 3: Number of addresses per collection that we collect.

| Name | Number | name | Number |
|---|---|---|---|
| DogePunks | 11,596 | Battle of BTC | 11,118 |
| BTC Virus | 9,993 | Bixels | 10,024 |
| Bitcoin Crypto DickButts | 8,195 | Mesh Beatles | 3,305 |
| OrdiRats | 2,317 | Bsos | 1,819 |
| 420 Rabbits | 1,192 | Block Gods | 1,049 |
| Taproot Cows | 557 | Pixel Panda Wars | 399 |
| STARBREEDER | 374 | bitCroSkull | 334 |
| Cubic A: Kaz Marquis | 196 | Familiar Fronks | 169 |
| Majo | 110 | Ordinal Cat Warriors | 102 |
| 10² Islands | 101 | iDclub Pass | 21 |

# B BITCOIN ORDINAL INSCRIPTION

## B.1 Detailed reasons for the effectiveness of the labeling method

Since taproot script spends can only be made from existing taproot outputs [46], inscriptions are created using a two-phase commit/reveal procedure. First, a taproot output is created in the commit transaction, committing to a script that contains the inscription content. Second, the reveal transaction spends the output of the commit transaction, revealing the inscription content in the blockchain.

To make an inscription a child of another, the parent inscription has to be inscribed and present in the wallet. Users use an inscription as an input in the reveal transaction, designating it as the parent of the new inscription. For a root to be acknowledged as the parent of a new inscription, the inputs of the reveal transaction should contain the parent inscription, thereby proving that the creator of the child inscription controls the parent inscription.

## B.2 Validation dataset

Our collection consists of 20 Bitcoin ordinal inscription collections, as shown in Table 2. These collections are created between February and May 2023, during the early phase of Bitcoin ordinal inscription development. The earliest collection *Familiar Fronts* in our dataset emerges in February 2023, with over two-thirds of these collections created from February to April 2023. Of these 20 collections, most contain hundreds to thousands of addresses, while a few contain about 10,000 addresses each.

In these collections, there is only one input address in reveal transactions for each ordinal inscription. Moreover, entities have the option to reuse addresses when creating ordinal inscriptions. For instance, the creator of the *iDclub Pass* collection uses 21 addresses to create 500 ordinal inscriptions.

