# OpenReview forum: "Exploring Unconfirmed Transactions for Effective Bitcoin Address Clustering"
_ACM.org/TheWebConf/2024/Conference — TheWebConf24 Oral_

### Official Review · Reviewer_rgyE · 2023-11-16

**Novelty:** 6
**Technical Quality:** 6

**Review:**

The authors of this paper present a novel set of heuristics that allows them to improve their ability to cluster addressed by virtue of including unconfirmed transactions. I thank the authors for their submission. While I am not an expert on this field, here are somethings that I think should be clarified in this paper:

- This paper relies on a modified version of Bitcoin Core, and the authors claim that 5 nodes are enough to collect all unconfirmed transactions. However, the authors do not explain how they know the total number of unconfirmed transactions? This needs to be made clear as it is central to the paper.
- There's a lack of discussion of what prior research used as a validation dataset and why the method proposed in 4.3 improves the SOTA.
- Based on Table 2, it is implied that your heuristic should be used in conjunction with previous work, this is not sufficiently clear.

This paper is well written and easy to follow, but as a minor detail that could improve readability, the authors can substitute the use of "his or her" and "he or she" by "their" and "them". This form of inclusive language is widely used and makes the text easier to read.

**Questions:**

- 3.2 Should introduce the concept of Taproot addresses. It is unclear why this is relevant to the paper right now.

**Reviewer Confidence:**

2: The reviewer is willing to defend the evaluation, but it is likely that the reviewer did not understand parts of the paper

**Scope:**

3: The work is somewhat relevant to the Web and to the track, and is of narrow interest to a sub-community

---

### Official Review · Reviewer_P4pE · 2023-11-20

**Novelty:** 6
**Technical Quality:** 6

**Review:**

The paper explores unconfirmed transactions in the mempool to advance Bitcoin address clustering, in contrast to the SOTA focusing only on the confirmed transactions in the confirmed blocks. The authors introduce a data collection framework and two novel clustering heuristics exploiting the specific behavior patterns in unconfirmed transactions, i.e., Replace-by-fee (RBF) and dependency chain. They construct a labeled dataset, publish the dataset, and experimentally show the effectiveness of their scheme by comparing with the SOTA clustering heuristics.

The evaluation and validation build on multiple SOTA schemes while comparing when the authors’ scheme and the unconfirmed transactions are used vs. not used. The analyses demonstrate the advantages of using their scheme, including the recall and precision performances and the number of identities additionally associated.

In addition to discussing the SOTA based on confirmed transactions, the authors discuss related research analyzing Bitcoin mempool and the unconfirmed transactions. The other research have different goals from this paper’s goal of address clustering.

**Questions:**

Does the UTP design and advantage come from having distributed set of nodes collecting the transaction information? Or does UTP further advance the unconfirmed transaction collections beyond that? In Figure 3, how does 0 (no UTP) compare with 1 (UTP but with one node)? It would also be good to show the results after the number of nodes exceed 5 to better support that the transaction number rarely increases beyond 5.

**Reviewer Confidence:**

4: The reviewer is certain that the evaluation is correct and very familiar with the relevant literature

**Scope:**

3: The work is somewhat relevant to the Web and to the track, and is of narrow interest to a sub-community

---

### Official Review · Reviewer_isLh · 2023-11-21

**Novelty:** 5
**Technical Quality:** 5

**Review:**

### Summary

This paper presents an innovative approach to Bitcoin address clustering, utilizing both confirmed and unconfirmed transactions—a novel exploration in the field. The introduction of two clustering heuristics, capitalizing on the replace-by-fee and unconfirmed transaction dependency chain mechanisms in unconfirmed transactions, distinguishes this work. The effectiveness of these heuristics is assessed through a comparative evaluation with six state-of-the-art clustering methods, employing precision, recall, weighted precision, and weighted recall metrics. The results demonstrate that the proposed clustering heuristics outperform state-of-the-art counterparts, revealing their capability to identify more entities and uncover additional address associations.

### Strengths
* Innovative method for clustering Bitcoin addresses, incorporating unconfirmed transactions.
* Comprehensive data collection encompassing both confirmed and unconfirmed transactions.
* Detailed comparison of six state-of-the-art clustering heuristics.

### Weaknesses
* Lack of clarity in the design of the *Heuristics Executor* component.
* Doubts about the representativeness and reliability of the labeled dataset.
* Questionable evaluation methodology, including an incomplete comparison with state-of-the-art clustering heuristics.

**Questions:**

In this paper, a novel clustering approach for Bitcoin addresses is introduced, utilizing both confirmed and unconfirmed transactions. However, certain aspects of the proposed approach raise questions regarding its feasibility and practicality.

First, the key component - *Heuristics Executor* - of the proposed approach lacks a clear explanation. While both confirmed and unconfirmed transactions are extracted and applied to the novel clustering heuristics, only the confirmed transactions are excluded from the application of state-of-the-art clustering heuristics. The rationale behind not applying unconfirmed transactions to the state-of-the-art clustering heuristics is not elucidated. Additionally, the final evaluation results appear to amalgamate outcomes from both the novel clustering heuristics and state-of-the-art clustering heuristics. Determining the optimal approach between these two remains unclear until the evaluation section, contributing to overall confusion in the presentation of the paper.

Second, the paper does not justify the representativeness and reliability of the labled dataset, as it may suffer from selection bias, noise, and ambiguity. Moreover, the paper does not report the size, distribution, and quality of this dataset, nor the methodology and criteria for labeling the addresses.

Third, the paper limits its analysis to only six state-of-the-art clustering heuristics, potentially resulting in an incomplete assessment of available methods. Furthermore, although precision appears satisfactory (exceeding 90%), the performance of the other three metrics, such as recall, is notably less impressive.

#### Minor Issues
* NU is not explained when it is first mentioned.
* Grammar issue in *On the one hand, a portion of unconfirmed transactions will inevitably turn into failed transactions*.

**Ethics Review Description:**

1. Bitcoin node monitoring for unconfirmed transaction collection 2. Address clustering for benign Bitcoin users

**Ethics Review Flag:**

Yes

**Reviewer Confidence:**

3: The reviewer is confident but not certain that the evaluation is correct

**Scope:**

2: The connection to the Web is incidental, e.g., use of Web data or API

---

### Official Review · Reviewer_6ksK · 2023-11-22

**Novelty:** 7
**Technical Quality:** 5

**Review:**

The paper discusses how to improve the effectiveness of Bitcoin address clustering by utilizing unconfirmed transactions in the transaction pool. The authors proposed two new clustering heuristics that respectively utilize the replacement mechanism and dependency chain mechanism in unconfirmed transactions to discover more address associations. This paper also proposes a reliable data collection framework and a labeling method based on Bitcoin ordinal inscriptions to verify clustering results. The authors’ experimental results show that their methods can significantly improve recall while maintaining high precision, with significant improvements over existing clustering heuristics.

Currently, heuristic clustering methods focus only on confirmed transactions, but unconfirmed transactions may be just as important for de-anonymizing Bitcoin users. Unconfirmed transactions can not only reflect the user's trading purpose, behavior and usage habits, but also reflect the information that cannot be provided in confirmed transactions.
This is the first study to use unconfirmed transactions to cluster Bitcoin addresses, providing a new idea for subsequent research on Bitcoin address clustering technology.

**Questions:**

Here are some suggestions for improving the paper:
1)	In section 3.3, the statement “a Bitcoin transaction is considered irreversible after the blockchain receives six new blocks.” is not explained clearly enough. What is the relationship between this transaction and the six new blocks? Is the transaction irreversible when it is added to the block and then six more blocks are added to the blockchain? Please describe this more specifically.
2)	In Section 3.3, two clustering heuristics in unconfirmed transactions are introduced, but their frequency in real-world environments has not been further validated in this study. It is recommended that the authors confirm the prevalence of these two mechanisms and the feasibility of the proposed clustering heuristics in real-world scenarios.

3)	The explanation of “ordinary inscriptions” is not very clear. A specific example can be used to explain the meaning of “ordinary inscriptions”, which will make it easier for readers to understand the significance of using ordinary inscriptions for database establishment.
4)	How much contribution do the two clustering methods proposed in this article contribute to the improvement of clustering effect? Or in other words, among unconfirmed transactions, is the replacement transaction more likely to occur or the dependent chain transaction more likely to occur? You can add some analysis in this area.


5)	Since the construction of annotated datasets based on Bitcoin ordinal inscriptions is a significant contribution, the paper should provide a specific introduction to the method of ordinal inscriptions labeling and the current proportion of transactions utilizing ordinal inscriptions. Additionally, a comparison between the experimental results using the annotated dataset and those without should be included.
6)	Table 2 shows relatively low weighted accuracy and weighted recall. The authors are encouraged to provide an explanation for these findings.
7)	In Section 4.5, Figure 7 demonstrates significant variations in experimental data before and after the appearance of the inscription. More comparative experiments before and after this time node should be conducted to elucidate the specific impact of the inscription.

**Reviewer Confidence:**

4: The reviewer is certain that the evaluation is correct and very familiar with the relevant literature

**Scope:**

4: The work is relevant to the Web and to the track, and is of broad interest to the community

---

### Official Review · Reviewer_9Hhh · 2023-11-30

**Novelty:** 4
**Technical Quality:** 4

**Review:**

First of all, I'd like to thank the authors to submit their paper to WWW'24. Although I am very unfamiliar with the topic, the paper was presented in such a way that (most of) it was comprehensible.

Strengths:
- While the fraction of unconfirmed transactions is only minimal (2.76%), it is still interesting to see that these can be used to reveal more address associations
- The proposed heuristics and usage of unconfirmed transactions has been adequately evaluated

Weaknesses:
- Unclear what the ultimate (practical) impact is on the ecosystem
- Evaluation focuses on the proposed methods, but doesn't go in-depth to evaluate the concrete impact

**Questions:**

- What is the *practical* impact of your proposed heuristics & using unconfirmed transactions?

**Ethics Review Description:**

No ethical concerns

**Reviewer Confidence:**

1: The reviewer's evaluation is an educated guess

**Scope:**

2: The connection to the Web is incidental, e.g., use of Web data or API

---

### Decision · Program_Chairs · 2024-01-22

**Decision:**

Accept (Oral)

**Comment:**

The paper addresses an important issue in clustering addresses on blockchains for various applications such as ransomware detection.
 One novelty of the paper is the usage of unconfirmed transactions. Mainly, previous work generally used the confirmed transactions only.
 This novel observation (i.e., unconfirmed transactions may help), I think would enable further research on the topic.

 ---